# Impact of and Coping with Post-Traumatic Symptoms of Refugees in Temporary Accommodations in Germany: A Qualitative Analysis

**DOI:** 10.3390/ijerph191710893

**Published:** 2022-09-01

**Authors:** Irja Rzepka, Catharina Zehetmair, Emma Roether, David Kindermann, Anna Cranz, Florian Junne, Hans-Christoph Friederich, Christoph Nikendei

**Affiliations:** 1Center for Psychosocial Medicine, Department for General Internal Medicine and Psychosomatics, Heidelberg University Hospital, 69120 Heidelberg, Germany; 2Department of Psychosomatic Medicine and Psychotherapy, Magdeburg University Hospital, 39120 Magdeburg, Germany

**Keywords:** refugees, traumatization, PTSD, coping, qualitative research

## Abstract

Due to pre-, peri- and post-migratory stress factors, refugees often experience higher levels of psychological stress than the general population. Post-traumatic stress disorder, in particular, has an increased prevalence in the refugee population. However, living conditions in the early post-migratory phase are characterized by many challenges. In the present qualitative study, 14 refugees with symptoms of PTSD from temporary accommodations in Germany were interviewed in semi-structured interviews about their individual experiences of the impact of their trauma sequelae symptoms on their current living conditions and interactions. Participants reported dealing with post-traumatic symptoms primarily through distraction strategies, such as working or learning the language or social interaction. Many reported a sense of mistrust as a result of traumatic experiences. Current stress factors cited included uncertain asylum status, worry about family members and lack of ability to influence living situations. The interactions between the post-traumatic symptoms and the living conditions of the refugees were highlighted. The effects of the symptomatology of trauma sequelae and the framework conditions under which refugees live can lead to aggravated psychological distress. Therefore, special attention must be paid to refugee mental health care.

## 1. Introduction

The civil war in Syria [1], ongoing conflicts in Afghanistan [2] and, most recently, the war in Ukraine [3] are just a few of the many conflicts around the world that have forced people to leave their homes in recent times. Around 100 million people worldwide are considered as refugees having fled their homes because of war, persecution and unstable living conditions [4]. In their home countries, during their flights to safe places, and after arriving at countries of refuge, they can experience stressful or traumatic events [5]. Refugees, therefore, belong to a population affected by an increased prevalence of mental illnesses, such as post-traumatic stress disorder (PTSD), depression and anxiety disorders, compared to the general population [6,7].

In particular, PTSD is a mental illness that can develop after experiencing an “extremely threatening or horrific event” [8] and affects, on average, 30% of the refugee population [6,7]. The condition impacts many aspects of life, especially participation in social life, and is associated with a lower quality of life [9]. After reaching destination countries, refugees face many further challenges: the process of asylum is often lengthy and difficult, and, during this time, their future perspectives are uncertain [10,11]. In addition, families can become separated, either during flight or due to restrictive immigration policies [12,13]. Family separation is one of the most distressing factors in the post-migratory phase [14]. In addition, there are other tasks to master, such as learning a new language and settling into a new society and culture. Numerous quantitative studies have examined the detrimental interaction between pre-migratory trauma and post-migratory stress [15,16,17,18]. However, variables cannot fully represent the complex situations of refugees who have been exposed to traumatic experiences and then have to face the challenges mentioned above [19]. To deepen existing knowledge by drawing on the subjective experiences of refugees [19], our aim was to more closely analyze the relationships between the effects of traumatic events and the challenges they face upon entering a country of refuge.

Therefore, in the present study, we interviewed 14 refugees with post-traumatic stress disorder about the impact of the traumatic events they experienced and their secondary symptoms on their everyday life in temporary accommodations in Germany, in addition to asking about their strategies for coping with these effects.

## 2. Materials and Methods

### 2.1. Participants and Study Design 

In February and March 2021, we conducted a descriptive study using qualitative, semi-structured interviews with participants housed in seven temporary accommodations in the Rhine-Neckar district (Baden-Wuerttemberg, Germany). The participants were either asylum seekers in the process of seeking asylum or authorized to stay in Germany with a temporary residence permit [20]. Inclusion criteria were an age of 18 or older, the ability to give informed consent and the ability to converse in German, English, French, Farsi, Arabic, Turkish or Serbian. 

### 2.2. Study Procedure

This study was part of a broader study within the temporary shelters in the Rhine-Neckar district. We had previously screened individuals who met the criteria for PTSD using screening questionnaires. We also used questionnaires to assess depression and anxiety [21]. Participants who had already been recruited for the previous project were informed via telephone about our current study and invited to participate in the interviews. If they agreed, we set face-to-face appointments for the interviews in their respective temporary accommodations. A verbal and written explanation of the study procedure, interview recording, pseudonymization and anonymization was provided to the participants. If they wanted to participate, they filled out an informed consent form. Then, the interview was conducted. If the participant could converse in German, English or French, the interview was conducted in whichever of these languages could be spoken. If an individual could not converse in one of the above languages, information about the study and the interview was communicated using a telephone interpreter. Before starting the study, all relevant documents (study information, chief investigator contact details and informed consent) were translated from German into English, French, Farsi, Arabic, Turkish and Serbian. 

### 2.3. Semi-Structured Qualitative Interviews 

Semi-structured, qualitative interviews were used to collect data regarding the interviewees’ experiences during their flights from their home countries, the traumatic events they experienced and how their symptoms impacted their daily lives regarding current stressors and resources in the post-migratory phase. Our research team developed the semi-structured interview guidelines. Methodological aspects of Helfferich [22] were followed. The semi-structured interviews comprised key questions focusing on the refugees’ daily lives, living in temporary accommodation, their professional lives and the integration process. The key questions were followed by probing and more detailed, clarifying questions. In order to avoid a one-sided presentation, these were designed to highlight the participants’ different perspectives on stressors and resources, depending on the course of the interview [23]. The interview guidelines used for this study are shown in Table A2 in the Appendix A.

### 2.4. Ethical Approval 

The Heidelberg University Ethics Committee approved the study (S-179/2021), and all participants gave their written informed consent per the Declaration of Helsinki (most recent version: Fortaleza, Brazil, 2013). 

### 2.5. Quantitative Data and Qualitative Data Analysis 

Demographic variables and baseline characteristics are presented using descriptive statistics via the Statistical Package for the Social Sciences (SPSS) program version 24 [24]. The qualitative face-to-face interviews were digitally recorded and transcribed verbatim by a research assistant using predefined transcription rules. The qualitative data were analyzed with the software MAXQDA [25], following the principles of qualitative inductive analysis described by Mayring [26]. Before examining the textual material, we defined the content-analytical unit of analysis as every statement (single or multiple sentences) referring to our key question. We reviewed the textual material of the transcribed interviews and identified single or a few content-bearing sentences as codes representing the most elemental units of meaning in each transcribed interview [27]. The interviews were processed sentence by sentence, and each unit of meaning was evaluated as to whether it contributed to answering the research question and whether an already created code could be applied to it. If this was not the case, a new code was created for the statement. The codes were labeled and summarized into relevant categories. After analyzing 50% of the textual material, we revised the categories and the whole coding system according to the logic of the previously defined categories. After that, we completed the analysis of the remaining textual material and grouped the categories into main themes [26]. Finally, the research team discussed the categories and main themes in detail and adjusted them if necessary. 

## 3. Results

### 3.1. Sample Characteristics

In total, *n* = 25 participants were asked to participate in the study. We interviewed *n* = 14 of 25 participants (56%). *n* = 5 (20%) participants declined participation due to time difficulties or lack of interest in the study, *n* = 5 (20%) did not respond to our request and *n* = 1 (4%) person had moved away from their temporary accommodation. Table 1 depicts the sample characteristics of the interviewees participating in the study. The interviews lasted between 20 and 51 min. *n* = 2 (14.3%) interviews were conducted in English and *n* = 1 (7.1%) interview was conducted in French and German. *n* = 10 (71.0%) interviews were conducted using a telephone interpreter (*n* = 7 in Farsi, *n* = 3 in Turkish). Regarding the reported symptoms, most of the participants stated forgetfulness or difficulties concentrating (*n* = 6), intrusive thoughts (*n* = 6) and “stress” (*n* = 6), as well as sleeping difficulties (*n* = 4) and ruminations (*n* = 3). Table A1 in the Appendix A shows all the reported symptoms of the participants.

### 3.2. Results of the Qualitative Interviews

We identified 220 single statements that were coded, paraphrased and summarized into categories. Finally, five categories could be derived. In the following paragraphs, we present the categories and example statements. Table A3 in the Appendix A shows examples for each category within the main themes. In addition, we have marked textual passages with an asterisk (*) where the statement had to be translated into English. 

(A)Traumatic events (18 codes)

In the interviews, we did not examine the traumatic events in detail in order to prevent re-traumatization [23]. However, throughout the interviews, the participants mentioned their traumatic experiences. Participants talked about physical violence against themselves. One participant reported having multiple wounds on his skin and how he had been burnt with a hot object and his fingernails had been pulled out. Two female participants reported sexual violence: one of them reported violence in her childhood, because “*women are less valuable in our culture**”, and another woman described ongoing sexual violence from her husband. Due to the lack of women’s rights in her country, she could not get a divorce. Other participants reported experiences of discrimination in their home country because they belonged to a specific cultural or religious minority or because of their sexual orientation. Another participant reported an explosion and being buried under ruins for hours afterwards. Furthermore, participants had also witnessed violence and detention. One participant said he would have been more cautious if he had known what his life would be like today and what he would experience.

(B)Dealing with traumatic experiences and following symptoms (31 codes)

The participants were asked how they dealt with their post-traumatic symptoms in their daily lives. Above all, distraction strategies were reported as ways of coping with symptoms, mainly negative or intrusive thoughts. Working (paid or voluntary work), learning German, going grocery shopping or finding distraction by surfing the internet were mentioned. One interviewee stated: “*I think too much but sometimes we can make me not to think too much is when I go to work. If I go to work, I interact with my work colleagues then this is making my mind to calm down, you understand. And in the work, you know, when I’m working, you know, I don’t have time. I’m busy doing things then. This is also making me not to think too much, you know about the stressful things*”. Another participant said, “*I try to use the time I spend learning German so that I don’t think about the difficult times I’ve had**”. These two quotations demonstrate how past events continued to influence the lives of these participants. Interviewees reported that social support from trustworthy people was essential in dealing with their symptoms. One woman explained: “*In total, I have two friends. One is in Frankfurt and lives with her boyfriend. And well, of course, she is still far away from me. And this friend in Mannheim is a bit, we have a better relationship. And the children of her calm me down, for example, I get a good feeling in any case. [‥] She knows everything, what problems I have, how I feel**”. One participant pointed out: “*We openly talk about things with our German friends. They have a listening ear. And they can listen to us and don’t judge us**”. On the other hand, interviewees frequently reported that there was no one they could open up to or trust. One man said, “*There is nobody to open your heart to and talk about the bad times or bad experiences you’ve had*”. Another person reported how he would try to calm himself down and “*not to think stupid things*”. However, he highlighted that there was nobody to talk about his pain or sorrow. Two interviewees said that they tried to keep things to themselves to avoid burdening others with their problems. One man stated instead that he was “*bottling everything up*”. Another participant related: “*With friends we sometimes speak about it. Maybe if we did not speak about it, we could more easily let it go or forget. But from time to time, we speak about it**”. These quotations suggest a sense of isolation in dealing with painful emotions. Furthermore, a few participants stated that they were seeking professional support or taking medication, primarily because of sleeping difficulties. Sporadically, participants reported wanting to seek psychotherapeutic help. However, at the time of the interview, none of the interviewees received any regular psychotherapy. This suggests that refugees do not seek external help until they have exhausted all their internal coping mechanisms. Furthermore, they highlighted the importance of the feeling of safety. Some interviewees mentioned that they hoped they would feel safe and secure in Germany. One female participant reported that the intrusive memories sometimes scared her and that she tried to calm herself by telling herself that she was safe in Germany. One woman said she felt safe because “*this person has no access to me. And therefore, I am sure here cannot happen anything to me**”. Furthermore, participants reported that they were glad to know that their children were in a safe country. In addition, interviewees highlighted hope, confidence and gratitude for their current situation as important feelings. One participant stated: “*All in all, we are very thankful that we are allowed to stay here. Nevertheless, my daughters are very successful; one makes her C1 German Class, the other daughter likes to paint and painted a very nice picture lately. And yeah, I try my best**”. Another statement was: “*We are very hopeful thanks to our nice friends and thanks to the love of God. So, we pray that everything will be all right. We are hopeful**“. Positive aspects of residency in Germany were named as helpful coping strategies.

(C)Changing views due to traumatic experiences (16 codes)

In the interviews, we asked how participants’ views of the world, fellow humans and themselves had changed after experiencing traumatic events. Multiple participants stated that they had become more suspicious of their environment and other people. Two female participants described that they had developed a feeling of mistrust against men. One of them said that, even though her current boyfriend was a nice and friendly man, she could not trust him. The other interviewee described her mistrust and even hatred of men. Another participant stated that he harbors a general distrust of people: “*for example, when I see people on the street, I’m thinking, okay, they are also capable of committing such a crime**”. Even though he said he knew that the laws in Germany deterred people from committing crime, this worry would always accompany him in his everyday life. One participant showed his feeling of mistrust against the recording of the interview because he was afraid that the information would be misused and would influence his asylum process. While some participants stated their mistrust against certain nationalities or men, others specified that their mistrust extended to people in general rather than to people of specific nationalities. One woman explained that she felt that danger was lurking everywhere, that “*it’s everywhere the same**”, no matter where she goes. A man described a feeling of hopelessness: “*I think at my present or my future like a normal person**”. One of the participants mentioned that as a consequence of the traumatic events she had experienced, she had reduced contact with people: “*Only today I opened the curtain. Before, I did not**”. 

(D)Current stress factors (83 codes)

In the interview, we asked about current stress factors in the participants’ everyday lives. The description of current stress factors took up a large part of the interview time. Participants frequently reported the asylum procedure and the legal uncertainty along with insecurities regarding the future, as stressful, contributing to a sense of hopelessness and loss of perspective. One interviewee stated that he felt that he was “*oscillating between the sky and the earth**”. Moreover, participants reported feeling very anxious about deportation back to their country of origin. One interviewee stated, “*If one day the decision comes that I have to go back to Turkey, I don’t want to live anymore. I want to live where I have rights, that others stay behind me and that I feel comfortable. And I totally don’t want to go back. All the time I live in this fear: Will I be sent back?**”. The existential fears that the participants had experienced due to the asylum procedure became very evident. Further, a mother described the impact that the uncertain situation had had on her daughter: “*My little daughter is currently so anxious that every time she goes to the toilet for two minutes, she fears that during her absence we could get deported. She cannot go to the toilet alone. [‥] The same problem occurred when she started going to school. Again and again, she felt so much fear about us not being home but deported when she came back from school* *”. The phase of waiting until the end of the asylum process often was described as “*wasted time**” and “*unbearable**”. Some participants highlighted that they were stressed because they did not understand the asylum procedure. One participant pointed out: “*And that in fact I did not understand: you have to leave the area, you cannot stay here, and if you go to another country, they will bring you back here**”. These examples highlight the severe burden of uncertain asylum status.

The majority of interviewees reported feeling burdened by worrying about other people. The worries concerned relatives or people close to them who had escaped from their home countries, as well as friends and family members who had stayed in their respective countries of origin under insecure living conditions. One statement regarding the separation from family members was: “*It makes me stress because of frustration, because of I don’t know what situation my family is, normally I have to be with my family. So, if your child is far from you, how can possibly your child really know the real father. It’s very difficult. Just to talk on the phone. It’s very sad*.”. One interviewee pointed out his worries and concerns about the well-being of people in his home country, who also belonged to a minority. He feared an “*end of the world*” because no one was protecting his people. This demonstrates that worries can extend beyond a person’s close family. Others worried about the mental health condition of loved ones or about the asylum procedures they had to undergo. One woman stated the effect that the separation from her children had had on her: “*Yes, I have a lot of stress. It’s because of so much stress from me. I’ve been away from my kids for six years, and that’s not little. [‥]. Yes, when I see the other children, my heart burns*.*”

Every interviewee highlighted that the ongoing process of asylum or uncertain asylum status impacted their daily life: they reported many challenges, such as searching for a flat: “*We’ve written over 500 emails and either they don’t respond or if we even get a viewing appointment and then when they hear what a situation we have, immediately balk and then they tell us off**.” Further, finding a job was challenging due to the need to obtain a work permit or the approval of certificates. One participant described the loss of status due to his flight: “*In my country of origin I had a job and a carrier. And here, for sure I have to start from zero. And to get from zero to one is difficult. That is really burdening for me**”. A flight might entail not just leaving one’s homeland but also losing personal achievements. Interviewees frequently depicted the monotony in their daily lives and the lack of opportunities to influence their circumstances. One exemplary statement was: “*Every day is a repetition of the other day. That is, there is no variety in the daily routine*”.* One interviewee stated: *“I used to solve my problems with logic and knowledge. But there are things which are not under my control. For instance, things related to the federal ministry for migration and refugees or other authorities. These things are not at all under my control, I don’t have any influence. That’s why these things give me a lot of worries and a lot of thoughts. I don’t have any influence on them**”. Furthermore, one significant emotion felt in daily life was a sense of hopelessness. One participant described: “*Throughout the day I am sleeping at home. I think I will die doing this. Because I don’t do something good to myself. I don’t learn anything new […] I am scared that every day in my life will be the same**”. These quotes point out the constraints to which asylum seekers are subject in these times. Further, limited financial circumstances hampered the interviewed participants in visiting their families regularly or finding a flat. Around half of the interviewees narrated that they had the feeling of little support or that they did not feel understood, for instance, regarding healthcare needs, finding a German language course, a workplace or a flat of their own, within the framework of the bureaucratic requirements.

(E)Experiences with integration in Germany (29 codes)

Regarding integration, both hampering and facilitating factors were mentioned by the participants. Most interviewees said that learning the language would be the key to integration. They pointed out that one can make contact with one’s environment through language. One statement was: “*The most important way to integrate in my opinion is language. So if you don’t know any language, don’t know German language, then you can’t exchange with these nice people**”. A few participants pointed out that they saw getting a job as the key factor in integration. This would make contacting other people more easy, solving financial issues and asylum procedure difficulties. One participant stated that “*at work, you get to know things, you talk to people… how to say, more and more integration**”. Some mentioned that it would be essential to adapt to the new way of life in Germany. The interviewees did not feel welcomed, which was a hindering factor. One interviewee noted that they had no sense of having arrived in German society or being welcomed gladly. Some mentioned cultural differences which would make it hard to get in contact with others and integrate. One interviewee pointed out: “*Yes, because of the culture. I [‥] grew up in completely different culture. At my age, it is sometimes difficult to adapt. And of course, for the German people, it is also sometimes difficult to accept us. And, therefore, I can say that I see the culture as an obstacle*.*” One participant stated that his traumatizing events would hamper his integration: “*Yes, of course, that also affects my integration. [‥] what you have experienced in your past, [‥] you can’t forget it completely. And if one could forget that, perhaps it would also be a disease. You would have to be maybe Alzheimer’s, then that would be possible. And yes, I am as I said 50–60% integrated at the moment. Would want to feel fully integrated of course. And yes, all that of course affects and delays also, delays also my integration**”. 

## 4. Discussion

This study aimed to shed light on the individual experience of traumatic events and PTSD symptoms among people who have fled their home countries, the impacts these experiences and symptoms have on their everyday lives, and the ways they have of dealing with them. Regarding the traumatic events reported, they mentioned different experiences, including physical, psychological and sexual violence, witnessing violence against other people and traumatizing events not due to human agency. Studies have shown that people forced to leave their countries often suffer many traumatic experiences [28]. On the one hand, determining factors can be located in the country of origin: human rights violations, handling of political opponents or minorities or war can increase the risk of experiencing traumatic events [29,30]. On the other hand, depending on the route, duration or stopping points along the way, the flight journey can include dangerous, traumatizing events [31,32]. These factors lead to the comparably high prevalence of mental illnesses in the population of refugees [6]. In the following, some aspects of the results will be emphasized and discussed in more detail.

As a result of traumatic experiences, the participants reported distressing symptoms individually, which affected them in their everyday lives. The participants in this sample named intrusions, concentration difficulties and a diffuse feeling of being stressed. The indefinable feeling of “being stressed”, “chronic stress” or “getting up stressed” that was repeatedly mentioned can be understood as a manifestation of post-traumatic hyperarousal or an inner tension in the context of a depression. However, the description of stress can also be attributed to the genuine challenges encountered in the post-migratory phase regardless of manifest mental illness. We will address these challenges in more detail in the discussion. 

Two coping strategies, in particular, were mentioned as ways of dealing with the consequences of traumatic experiences: first, distraction strategies, such as learning German or working, were described as helpful ways of handling post-traumatic symptoms. In Germany, refugees are obliged to participate in so-called “integration courses”, in which the necessary language skills are taught. These courses are usually full-time for at least half a year [33]. “Studying German” is therefore often a daily duty and part of a structured everyday life for people seeking asylum in Germany, especially in the early post-migratory phase. Language skills can be enhanced by improving post-traumatic symptoms and general well-being [34]. Interviewees also frequently mentioned difficulty concentrating in connection with learning the language. Poor concentration can be understood as a symptom of PTSD and depression [35,36]. Concentration difficulties as symptoms of mental illness can hinder learning a new language [37], which can also lead to frustration and worsened mental health status [38]. Second, working was mentioned as another distraction strategy and as a tool for making contact with people, learning the language and receiving appreciation. However, the possibility of finding work is often limited because of legal prerequisites, such as obtaining a work permit [39].

Regarding social support as a handling strategy, participants emphasized contact with people close to them as a resource. Several interviewees also mentioned that social support would be desirable for dealing with their psychological distress but pointed out that they had no one to talk to or trust. Research has shown that patients with more social support reported less severe PTSD symptoms [40]. However, the social network can be limited by anxiety disorders, such as PTSD, through emotional numbness [41,42], while depression can also lead to reduced social contact through social withdrawal [43]. In a study by Aarethun, Sandal [44] with Syrian refugees, another person’s trustworthiness was one of the most critical aspects of social contact. The participants repeatedly named mistrust as a consequence of traumatic events, which has also been confirmed in quantitative studies [45,46]. The need for trustworthiness and high levels of mistrust or social withdrawal can hinder establishing contact with new people. However, due to stigmatization, members of the host society may likewise harbor mistrust towards those who have immigrated to the country [47]. Furthermore, the general living conditions of refugees, for instance, in temporary accommodations, which are often isolated and poorly connected, can hinder the establishment of social contacts [48]. Losing their own existing social networks can also make it challenging to establish new trusting contacts and use the social network as a resource [49]. Moreover, one of the stated stress-inducing factors was worry about family members, due to separation from them, or about their general well-being in new living situations. In the case of refugees, the family, as a part of their original social network, can be a stabilizing factor with respect to their mental health [50], and the grief caused by family separation is one of the most burdensome factors in the post-migratory phase [14]. Beyond that, it can correlate with more severe PTSD, depression and anxiety symptoms, for example, due to ongoing impending danger to loved ones, even though personal safety is assured [17]. Thus, not only does the family cease to be a resource but caring for the family becomes an additional stressful factor, when social support is needed the most.

Furthermore, almost all of the participants mentioned the uncertainty of the asylum process and its consequences for employment, housing or future planning as significant stress factors, concomitant with the limited possibilities of influencing their situations. These challenges, presented in the Results section, were expressed by different interview participants. However, it is not unlikely that refugees in the post-migratory phase must deal with several of these challenges at the same time. A lengthy asylum process is well known to be correlated with worsened mental health status [51]. Beyond that, feelings of powerlessness and being at the mercy of someone are emotions that have often been associated with traumatizing events [52]. In the living situation of a person seeking protection in a new country, this takes on a broader meaning, since the decision-making power concerning one’s future is in the hands of an authority and there is the possible danger of being sent back to one’s home country, where traumatic events may have occurred. A sense of powerlessness and the fundamental uncertainty and lack of decision-making influence over their own future can be a double burden for refugees with PTSD.

In the last part of the interview, participants were asked about their experiences of integration in Germany. The participants listed language skills and a workplace as prerequisites for integration into German society. These two aspects and the associated difficulties have been highlighted concerning the handling of trauma symptoms. Regarding mental health, it is known that limited access to the labor market and consequent inactivity can contribute to deterioration in mental health status [53,54]. Acquiring sufficient language skills, in turn, promotes the likelihood of integration into the working environment [55]. Lastly, cultural differences, possible discrimination and uncertainty during the process of asylum were described as hindering factors in relation to integrating into the new country. The acculturation process, balancing maintaining one’s own culture and participating in the new country’s culture [56], can exacerbate PTSD and depression symptoms [57]. Cultural differences can also lead to experiences of discrimination based on origin, asylum status or religion, and further exacerbate psychological distress. Furthermore, so-called microaggressions can impair mental health status [58]. The difficulties of making contact with people in a new culture due to language barriers, perceived mistrust of people and uncertainty about one’s future in the country of refuge can further complicate the process.

The results of the current study show that the effects of traumatic sequelae in refugees and their living situations under insecure residency conditions reinforce each other and lead to a double burden of psychological distress. Moreover, access to therapeutic care for refugees is limited [59] and poses several challenges [60]. Therefore, it is necessary to provide adequate support for traumatized refugees and thus facilitate their integration into society.

## 5. Limitations

In this study, 14 individuals forced to leave their country were interviewed about how PTSD has affected their daily lives and how they cope with their symptoms. Some limitations need to be indicated for the results to be appropriately interpreted. A sample size of 14 participants from a heterogenous population can be considered small. However, according to Hennink and Kaiser (2022) [61], data saturation in qualitative research can be reached on average with 9–17 interviews. Apart from this, we did not use a controlled study design. Refugees who were not experiencing post-traumatic stress could have been included as a control group for purposes of comparison and to refine the problems identified. Furthermore, participants were screened using screening questionnaires (PC-PTSD-5, PHQ-4) in our first sub-project [62,63]. However, screening questionnaires cannot replace a standardized clinical interview for a definite diagnosis. Therefore, insights into individual experiences of symptoms of traumatic sequelae under conditions of undetermined and uncertain residency status in Germany have been presented. However, these would have been influenced by the diversity of the refugees’ histories, the specific post-migration living conditions in Germany, and the situational dynamics and expectations of the researchers and participants [23]. As a result, the results, though they can paint a vivid picture of the refugees’ experiences, are not necessarily representative of all refugees. For 10 out of 14 interviews, a telephone interpreter was consulted. Using a telephone interpreter can lead to informational losses in terms of precise formulations and descriptions of impressions due to triangulation [64]. Coincidentally, however, the same Persian interpreter was engaged to interpret for five of the participants. As a result, the interpreter was already familiar with the interview questions and the study framework of the study by the second interview, which made the subsequent interviews easier to conduct. It must be mentioned that the interviews were conducted in Germany. Thus, the refugees were living under specific national regulations [20,39,65] and circumstances that influenced their living situations. Legal regulations for refugees, however, differ between countries and consequently lead to differences in living conditions. Lastly, it should be mentioned that the interviews were conducted during the COVID-19 pandemic, which influenced the refugees’ living situations and mental health [66], even though this was not explicitly targeted in the interview.

## 6. Conclusions

Refugees suffering from symptoms of post-traumatic stress disorder reported distraction as their primary coping strategy, for example, learning the language, working or social contact. Considerable mistrust was repeatedly named as a consequence of the experience of traumatic symptoms. This can hinder the establishment of trustful relationships, while, in turn, social contact was described by the participants as helpful in dealing with their symptomatology. Uncertain residency status is a significant strain. Against a background of post-traumatic stress disorder, this can become a persistent existential threat. Living conditions in the post-migratory phase are challenging for people seeking protection in their countries of refuge. The participants described negative interactions between traumatizing past experiences and living conditions while under the influence of both during the asylum process and during the asylum process. Traumatized refugees under uncertain residency conditions carry a double burden of psychological distress. Special attention to the therapeutic care of refugees is needed to facilitate their integration. 

## Figures and Tables

**Table 1 ijerph-19-10893-t001:** Sociodemographic data of the participants.

Sociodemographic Data		
Age		
Mean	36.1	
SD	10.2	
Min./Max.	22/53	
Gender	*n*	%
Male	8	54.1%
Female	6	42.9%
Country of origin	*n*	%
Iran	7	50.0%
Turkey	2	14.3%
The Gambia	2	14.3%
China	1	7.1%
Morocco	1	7.1%
Armenia	1	7.1%
Years in Germany	*n*	%
2015	2	14.3%
2017	2	14.3%
2018	5	35.7%
2019	4	28.6%
Not specified	1	7.1%
Asylum status	*n*	%
Asylum received	1	7.1%
Awaiting asylum decision	6	42.9%
Received tolerance (German: ‘Duldung’)	5	35.7%
Not specified	2	14.3%
Screening data	M (SD); Range	
PC-PTSD-5	4.14 (0.86); 3–5	
PHQ-2	4.25 (1.71); 1–6	
GAD-2	4.25 (1.71); 1–6	

## Data Availability

The datasets used and analyzed during the current study are available from the corresponding author upon reasonable request.

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
