# Peer review of "Impact of and Coping with Post-Traumatic Symptoms of Refugees in Temporary Accommodations in Germany: A Qualitative Analysis"

_ijerph, 2022, doi:10.3390/ijerph191710893_

Round 1
Reviewer 1 Report
In this qualitative study, 14 refugees with symptoms of PTSD from temporary shelters in Germany were interviewed. Individual experiences were recorded. Refugees reported overcoming stress via distraction strategies of working, learning local language or social interactions. Several stressful factors were noted in the cohort including immigration status uncertainty, worry about family members, and helplessness. While the study is important, and topical, enthusiasm is lessened from a very small sample size, lack of appropriate control group, and robust statistical analysis. There also is limited innovation or novelty in work as several studies already have reported the connection between displacement of refugees with aggravated psychological distress.
Author Response
Please see the attachement

Reviewer 2 Report
The authors focus on the impact of and coping with post-traumatic symptoms of traumatized refugees. The most probably it would be good to mention in the paper’s title that the research was settled in German context. The problem touched by the authors is very topical and prospective in terms of research. However, the study was carried out only on a sample of 14 respondents, which means that they are of a exploratory nature, and their generalization would be a mistake (the authors rightly mention this limitation and other limitations of their research). Certainly, this paper is a good starting point for more thorough and broader future research in this field.
Authors’ English is correct. The paper requires some editorial/technical amendments but not many linguistic corrections.
The research aim was clearly stated. As to the research method, the authors mention in the title that it is a qualitative analysis but in the item 2.5 they conduct quantitative and qualitative data analysis. So this aspect should be dealt with more consistently.
The literature presented in the references is relevant and up-to-date and the sources are rich; however, there are some editorial errors or omissions and appropriate complements are required:
– The source ‘Nesterko, Y., et al.’ in the references (item no. 17) was not published in the year 2020 but in 2019.
– The source ‘Ozer, E.J., et al.’ in the references (item no. 29), should include the page range ‘52–73’.
– The source ‘Walther, L., et al.’ in the references (item no. 42), should be completed with the page ‘p. 576481’.
– In case of the source ‘Pinzón-Espinosa, J., et al.’ (item no. 55), the issue number should be written ‘280(Pt A)’ instead of ‘280’ (Pt A = Part A).
The results are presented in a clear way and the discussion conducted correctly, however the conclusions are quite lapidary. It also seems that the Introduction section should be a bit longer.
Author Response
Please see the attachement

Reviewer 3 Report
Thank you for your thoughtful research project about the mental health challenges of refugee populations. I do not have any problems with the research method or the project in general--I think it is very important to identify helpful approaches to mental health obstacles of refugee/asylum seeking populations in third/resettlement countries, so I applaud your emphasis. I have some points of disagreement as to how the research material is presented:
1) The article refers to "Traumatized refugees" as if they are a specific category of refugees, even though it is clear from your methodology that you did not necessarily seek out refugees with specific types of traumatic experiences such as rape, torture or specific injury. In my experience of fieldwork, the entire refugee journey is traumatic, and every refugee and asylum seeker is traumatized to a certain extent. I have not met an untraumatized refugee. Even the definition of refugee, in the 1951 Convention, includes "well-founded fear" as its basis, which is part of the experience of trauma. So, I would refer to the refugees somewhat differently, not as "Traumatized refugees."
2) Articles written on mental health of refugee populations sometimes contribute to the medicalization of refugees, as if they are damaged goods from the outset, that they suffer more from depression and other disorders compared to the populations in the host society. Therefore, it is important to de-securitize and re-humanize refugees as we recognize their mental health struggles. Throughout the article, I felt like the refugees' words were being used to "prove" that they were depressed, or suffered from anxiety disorders or other ailments. Wouldn't anyone separated from their families worry about their families? Wouldn't anyone without legal status in a country worry about deportation? Are refugees necessarily more "depressed" than veterans, homeless communities, children in foster care systems, prisoners in jail? I would love to see a little more awareness of the stigmatization of refugees through medicalization and an effort to avoid this pitfall in the writing of this article.
3) The article uses the word refugee as a general term, for people who have applied for asylum in the German legal system. The problems facing these populations are very different than a UNHCR recognized refugee who is being resettled in Germany or another country, as they would get the legal protections emanating from refugee status and not have the stressors of deportation and legal work. I think it is important to realize that the asylum systems, specifically, are more conducive to generating anxiety and stress due to the ambiguities and negative outcomes associated with them. In that sense, on top of all the trauma that the refugee has had to endure on the way to Germany, following the trauma of fleeing from their country, the defensive legal systems in which we place them in developed countries "add" further stressors to their lives that can serve as the straw that breaks the camel's back. I think this was, indeed, one of your findings. So, I would like to see revisions that don't use the term "refugee" so generically, generalizing the terrible experiences of asylum seekers as typical of refugees, excusing how our legal systems that try to defend our countries are responsible for creating new forms of bureaucratized trauma that may be psychologically more harmful than all that has happen to that person up until that point. I think it is important to underline that research on post-resettlement trauma shows that it has an impact disproportional to the trauma experienced along the way to the host country.
4) Various sections "list" refugee sentences back to back, with no commentary. I think this presentation of the data does not do justice, because there are sentences or themes there that are worth writing a few more words about, underscoring, or explaining. I would prefer to see a little more analysis along the way, as a qualitative study.
Author Response
Please see the attachement

Round 2
Reviewer 1 Report
The authors have addressed reviewer comments in a diligent manner. The concern of sample size also is addressed.
Author Response
Please see the attachement
